# Suicidal Behaviour, including Ideation and Self-Harm, in Young Migrants: A Systematic Review

**DOI:** 10.3390/ijerph19148329

**Published:** 2022-07-07

**Authors:** Aditya Basu, Alexandra Boland, Katrina Witt, Jo Robinson

**Affiliations:** 1Orygen, Parkville, VIC 3052, Australia; alexandra.boland@umassmed.edu (A.B.); katrina.witt@orygen.org.au (K.W.); jo.robinson@orygen.org.au (J.R.); 2Centre for Youth Mental Health, The University of Melbourne, Parkville, VIC 3052, Australia; 3Department of Family Medicine & Community Health, University of Massachusetts Medical School, Worcester, MA 01655, USA

**Keywords:** suicide, suicidal behaviour, self-harm, young migrants, systematic review

## Abstract

Young people experience high rates of suicidal ideation, self-harm, suicide attempt and death due to suicide. As a result of increasing globalisation, young people are increasingly mobile and can migrate from one country to another seeking educational and employment opportunities. With a growing number of young migrants, it is important to understand the prevalence of suicidal behaviour among this population group. We systematically searched Medline, Embase, and PsycINFO from inception until 31 March 2022. Eligible studies were those providing data on suicidal ideation, self-harm, suicide attempt, and death due to suicide. Seventeen studies were included in the review, some of which provided data on multiple outcomes of interest. Twelve studies provided data on suicidal ideation, five provided data on self-harm, eight provided data on suicide attempt, and one study had data on suicide death among young migrants. The quality of the included studies was varied and limited. The studies included in this review commonly reported that young migrants experience higher rates of self-harm and suicide attempt, but no major differences in suicidal ideation and suicide death compared to non-migrant young people. However, the limited number of studies focused on suicidal behaviour among young migrants highlights the need for further high-quality studies to capture accurate information. This will enable the development of policies and interventions that reduce the risk of suicidal behaviour among young migrants.

## 1. Introduction

Young people experience high rates of suicidal ideation (defined as “thoughts of engaging in behaviour intended to end one’s life” [1]). They also engage in self-harm (defined as intentional self-poisoning or self-injury, irrespective of motive or the extent of suicidal intent [1]) and suicide attempt (defined as “engagement in potentially self-injurious behaviour with at least some intent to die” [1]). Research has indicated that young people also have higher rates of death due to suicide [2,3,4,5]. Previous suicide attempt and suicidal ideation are the greatest predictors of future suicide risk [6,7,8,9,10]. Globally, evidence indicates that between 45 and 60% of adolescents have experienced suicidal thoughts at least once in their lifetime [11,12]. It is estimated that between 14 and 25% of young people will engage in self-harm at some time in their life, usually commencing from adolescence [10]. The overall rate of suicide attempts among young people is three times higher than the rate among adults over 30 years of age [12,13,14]. Suicide is defined as a fatal self-injurious act with some evidence of intent to die [15]. Globally among those aged between 10 to 24 years, suicide is the second most common cause of death, after road traffic accidents. It is estimated that approximately 164,000 self-inflicted deaths occur among people aged under 25 years [16]. Most suicide deaths among young people occur within the age of 15–24 years [17]. 

Migrants may be at a higher risk of suicidal ideation, suicide attempts, self-harm, and death by suicide compared to the general population [18,19,20]. Suicide risk may vary among ethnic minorities, and they may experience different risk factors for suicidal behaviour compared to the native population of a country [21,22,23]. Some evidence indicates that acculturation, a process by which individuals acquire the attitudes, values, customs, beliefs, and behaviours of a different culture, may in fact increase the risk of suicidal behaviour among some migrants [24]. Research has indicated that suicidal behaviour and immigration may be associated with one another due to acculturation stress, changes in social roles and socioeconomic status, experiencing discrimination and social marginalisation in the host country, as well as feelings of isolation and loneliness [25,26,27]. This may be particularly salient for young migrants due to unique developmental factors and the stress of assimilating into the host country’s culture and forming new social connections, while retaining elements of their original culture and maintaining bonds with existing family and friends [28]. Overall, this indicates that a variety of potential factors, before, during, and after migration, can increase the risk of suicidal behaviour among migrants. 

Commonly, reviews evaluate suicidal behaviour among young people [11,29,30,31] or among migrants [30,31,32,33,34], but not both. Despite the myriad of risk factors, the rates of suicidal behaviour, including suicidal ideation, self-harm, suicide attempt, and suicide death among young migrants is not well understood. It is possible that young migrants face a dual burden that increases the risk of suicidal behaviour because they are affected by the risks due to migration as well as the risks due to young age. Understanding these factors is particularly critical in the modern age, where levels of migration are predicted to increase with people aiming to move to other countries for better employment and education opportunities and leave behind conflict and climate emergencies [34]. The aim of this review is to synthesise the evidence regarding the rates of suicide, attempted suicide, suicide ideation, and self-harm among young migrants. Young people are defined, as per the Lancet Commission on Adolescent Health and Wellbeing’s definition of “young people”, as those aged between 10 and 24 years [10]. Although there is no standardised legal definition of an international migrant, migrant status is defined in this review as per the International Organization for Migration (IOM), as any person who is moving or has moved across an international border or within a state away from their habitual place of residence regardless of the person’s legal status; whether the movement is voluntary or involuntary; what the causes for the movement are; or what the length of the stay is [35].

## 2. Materials and Methods

The methodology was informed by the Preferred Reporting Items for Systematic Reviews and Meta-Analyses (PRISMA) guidelines [36]. The review was pre-registered on PROSPERO, with the registration number CRD42019148233. 

### 2.1. Inclusion Criteria 

Studies were included in this review if they: (1) presented the findings of peer-reviewed research using any of the following study designs: case studies, interrupted time series studies, cohort studies, cross-sectional surveys, and before and after studies; (2) reported on the incidence and/or prevalence of suicidal ideation, self-harm, attempted suicide, and/or suicide among young migrants. The review included studies from all over the world. 

There were no limitations placed on how the studies collected data, which means studies that focused on objectively recorded data (e.g., hospital data, death records, etc.) and those that focused on subjectively reported data (e.g., self-reported or collateral informant reports, etc.) were included. Data on the proportion of individuals that had experienced suicidal ideation, self-harm, attempted suicide, and/or suicide death had to be provided (either in the papers or following correspondence with the study authors). Studies were also included in this review if: (3) they focused on young migrants aged between 10 and 24 years (inclusive) who had migrated between one country and another at any point in their lives. Studies were also included if some participants were over or under the age limit, but the mean or median age of the participants was within the age range of 10–24 years. 

The primary outcome of this review was self-harm, which could be ascertained either from objective sources (e.g., hospital, clinical, and/or medical records or from population registries) or from self-reported sources (e.g., family, peers, and/or neighbours). The secondary outcomes of this review were: (1) suicidal ideation, (2) suicide attempt, and (3) suicide death, all of which could be ascertained either from objective sources (e.g., hospital, clinical, and/or medical records or from population registries) or from self-reported sources (e.g., family, peers, and/or neighbours). 

### 2.2. Search Strategy 

We searched Medline, Embase, and PsycINFO from inception until 31 March 2022. Keywords relevant to suicidal behaviour and young migrants were combined using standard Boolean operators. Key words were developed by consensus among the author group. In addition, we hand-searched the reference lists of all previous systematic reviews retrieved via the search. Further detail on the electronic search strategy is available in Appendix B. 

The primary author (A Basu) conducted the searches, combined records, and removed duplicates. In the first instance, study titles and abstracts were screened by two review authors (A Basu and A Boland). Discrepancies were resolved by discussion. In the second stage of screening, full texts of potentially relevant studies were screened for inclusion by the same two review authors. 

### 2.3. Data Extraction

Data were extracted independently by A Basu and A Boland using a pro forma, which was pilot tested on a random sample of 20% of the records. The following information was extracted: (i) author(s); (ii) publication year; (iii) sample size; (iv) country/region of birth; (v) country of destination; (vi) study design; (vii) source of study participants; and (viii) outcome data on suicidal ideation, self-harm, attempted suicide, and/or suicide death. 

### 2.4. Study Quality

Study quality was assessed using the National Heart, Lung, and Blood Institute’s Quality Assessment Tool for Observation Cohort and Cross-Sectional Studies [37]. Each study was independently rated by two reviewers (A Basu and A Boland). 

The tool contains 14 criteria to determine the quality of each study and comprises several items affecting external validity (e.g., clearly articulated objectives, study population(s), adequate participation rates, pre-specified and consistently applied inclusion/exclusion criteria) designed to ensure the study achieves a true or close representation of the target population, and a justification of the sample size. Further items assess factors affecting the internal validity of the study (e.g., ensuring measurement of the exposure of interest [i.e., migration] occurred prior to measurement of the outcome [i.e., suicidal ideation, self-harm, attempted suicide and/or suicide death]; ensuring a sufficient time has elapsed between the exposure of interest and the outcome(s) of interest to ensure an association could be determined; methods of outcome ascertainment that are robust, valid, reliable, and applied consistently for all participants; and a loss to follow-up of no greater than 20%). 

Each criterion was rated as either yes, no, or “other” (that is, cannot determine, not reported, or not applicable), and an overall rating for each study was reached as either: “good,” “fair,” or “poor”. A study was determined to meet a study quality criterion if both authors agreed the criterion was present, and a study quality criterion was judged as not met if at least one author rater determined that the criterion was absent. Additionally, study quality was rated as “unclear” or “cannot determine” if insufficient information was reported to make a judgment. No allocation blinding occurred as this was not possible and that exposures (immigration) would have always occurred before outcomes were measured. 

### 2.5. Data Synthesis

A narrative synthesis of the results was undertaken to obtain insights from the data. Meta-analysis was not possible due to the heterogeneity in the outcome measures used between studies and the variety of settings from which data were captured, limiting the ability to accurately compare results. 

## 3. Results

### 3.1. Search Results

The search strategy identified 3209 studies; a further 18 were identified through reviewing the reference lists of related literature, giving a total of 3227 studies identified. After duplicates were removed, the remaining 1274 studies were screened based on their title. Of these, 1229 studies were excluded because they did not meet the criteria for inclusion. The remaining 45 studies were then screened for eligibility based on their full text. Twenty-eight were excluded because the authors were unable to determine the age of the studied population or the exact number of immigrants in the study participants. A total of 17 studies were included in this review. Figure 1 contains a visual overview of this selection process, while Table 1 contains the characteristics of included studies. 

### 3.2. Overall Description of Included Studies 

The 13 included studies were published between 1999 and 2019, with the majority (n = 9) published after 2011. Almost half of the studies (n = 6) recruited participants from educational settings—either high school or university. Five studies recruited participants from the general population, while one recruited participants from a labour camp and another from mental health clinics. Seven studies used a survey to collect data, while the remaining six studies analyzed data from population-based registries. None of the studies included information on comorbid psychiatric conditions or diagnosed mental illness for all participants. Although none of the studies included a specific category looking at the reason for migration, it was possible to deduce that participants in one study migrated for educational opportunities (international students at university) [39], participants in another study migrated for employment opportunities (migrant workers) [38]; and participants in two studies migrated as refugees or asylum seekers [40,54].

### 3.3. Study Quality 

The National Health Lung and Blood Institute’s Quality Assessment Tool for Observation Cohort and Cross-Sectional Studies was used to assess the risk of bias in each of the included studies. This tool provided an appropriate framework with which to evaluate the risk of bias for this topic. All the studies in this review were rated as either “fair” or “poor” study quality according to the tool. Most studies did not use robust controls to limit the risk of bias whenever possible and relied on convenience sampling methods instead. Of the final score given to the studies, 5 had a score of “fair” [38,40,41,42,48] and 12 had a score of “poor” [39,43,44,45,46,47,49,50,51,52,53,54]. None of the studies obtained a “good” score. Ratings for all 17 studies are available in the Appendix A provided.

### 3.4. Outcomes

#### 3.4.1. Suicidal Ideation

Twelve studies reported data on suicidal ideation, all of which collected data from surveys of relevant population groups [39,40,41,42,43,46,47,49,50,51,53]. Eight of these 12 studies compared the results of migrants with non-migrants with mixed findings [39,41,42,46,47,49,50,53].

Three studies found that non-migrant young people have higher rates of suicidal ideation compared to young migrants [39,41,47]. The first of these reported the results of a survey of Canadian tertiary education students, comparing those of Chinese origin with others [39]. The study found that 35% of Chinese Canadians had suicidal thoughts, compared to 45% of non-Chinese origin. No differences were found between males and females. No information was provided about confounding variables, such as pre-existing history of mental illness and suicidal behaviour, time spent in Canada, or socioeconomic status. The second study analysed the findings of the 2006 Boston Youth Survey (BYS), which collected data from selected public high school students in Boston, U.S. [41] The study broke down the analysis into various sub-groups. It found that of foreign-born students who had spent less than 4 years in the country, 2.7% reported having suicidal thoughts, while 9.6% of foreign-born students who had lived in the country for longer than 4 years reported suicidal thoughts. The study also broke the students down into those with Hispanic backgrounds, and those who did not have Hispanic backgrounds. When looking specifically at non-Hispanic students, the study found that 8.8% of non-migrants reported suicidal ideation compared to 5.9% of migrants. The differences were reversed and less pronounced when looking at Hispanic students—9.7% of non-migrants reported suicidal ideation compared to 10.4% of migrants. The third study used data from the National Longitudinal Study of Adolescent to Adult Health (Add Health), a nationally representative study of U.S. adolescents, with the population broken up into three groups—adolescents, early young adults, and young adults [47]. The main population of interest was Asian Americans, that is, individuals who had either migrated to the U.S. from Asian countries or had parents who had migrated to the U.S. from Asia. It found that between 3.5–14.5% of Asian migrants who speak English at home experienced suicidal ideation, while 4.6–9% of Asian migrants who do not speak English at home experienced suicidal ideation. Of people who were born in the U.S. but had parents from Asia, 1.0–19.2% experienced suicidal ideation. The non-migrant comparison group was non-Hispanic whites, 7.3–14.2% of who reported suicidal ideation. None of the three studies provided any further information about other confounding variables, such as previous diagnosis of mental illness, socioeconomic status prior to and after migration, and other such factors. 

Conversely, three studies found that migrants report higher levels of suicidal ideation than non-migrants [42,46,49]. One study found that 37.4% of young migrants reported suicidal ideation, while 35.9% of non-migrants did [42]. While young migrants from Poland, North-/West Europe, and Southern Europe reported a higher lifetime prevalence of suicidal thoughts (47.5%, 42.3% and 39.3%, respectively) than adolescents without a migration background (36%), the highest lifetime prevalence of suicidal ideation was reported by students with an Asian migration background (49.5%). Another study presented the findings of a school-based survey of European adolescents as part of the Saving and Empowering Young Lives in Europe (SEYLE) study [46]. This study found that 5.7% of young migrants from Europe and 6.7% of non-European migrants had serious suicidal ideation, compared to 3.2% of non-migrants. There was no detailed information reported regarding country of birth. Furthermore, no further information was available to determine if factors such as gender, age of migration, experience of traumatic events, or mental illness or any other factors influenced suicidal ideation. The third study reported findings from a survey of Russian-born migrants to Israel aged between 11 and 18 years, comparing the findings to an equivalent age group of non-migrants in Russia and Israel [49]. The study found that 10.9% of young migrants reported suicidal ideation compared to between 3.5% and 8.7% non-migrants. Of those reporting suicidal ideation, 2.5% did so “very frequently”, while 1% of Russian non-migrants and 5.8% Israeli non-migrants did. The study also found that those reporting intense suicidal ideation were more likely to have immigrated at an earlier age than those who rarely thought about suicide. The study reported no gender differences between male and female immigrants. 

One study reported mixed results, with 38.1% of young migrants from Turkey reporting suicidal ideation, followed by 17.9% of non-migrant Dutch young people, and the lowest reported by young Moroccan migrants (12.8%) [53]. Similarly, another study found that there were similar rates of suicidal ideation among young migrant children compared to non-migrant children, with 24% of young migrants reporting suicidal thoughts compared to 26% of non-migrants [50]. The study looked solely at young migrant refugees from Bosnia and Herzegovina displaced by war and living in Slovenia, compared to non-migrants born in Slovenia. No information was available about previous diagnosis of mental illness, historical suicidal behaviour, and socioeconomic status prior to and after migration among other confounding factors. Four studies had no control groups and presented data for young migrants only, meaning the data could not be compared to those of non-migrants [38,40,43,51]. Rates of suicidal ideation ranged from 11.2% [43] to 94% [51]. 

#### 3.4.2. Self-Harm

Five studies reported data on self-harm, out of which three studies provided data for migrants and non-migrants [41,42,48], while two provided data on migrants only [51,54]. Data from all studies were obtained through self-reported episodes of self-harm. 

One study found that there was no difference between non-Hispanic migrants and non-migrants, with 7.5% of both groups having engaged in self-harm. Of young Hispanics born in the US, 6.5% engaged in self-harm, compared to 11.5% of young Hispanic migrants [41]. Data for this study came from analysing the results of the 2006 Boston Youth Survey (BYS). The study did not provide a breakdown of the country of origin, instead grouping the findings into U.S.-born and non-U.S.-born young people, as well as breaking down the data into Hispanic and non-Hispanic ethnic groups. No information of sex, psychological health status, socioeconomic status, or any other findings were reported. Another study reported on a survey of school students in the state of Lower Saxony in Germany and found that 19.6% of migrants engaged in self-harm in a 12-month period, compared to 17.2% of non-migrants [42]. The study identified the country that some migrants were born in (Turkey and Poland), while other migrants were grouped into regions of origin (Former Soviet Union, Former Yugoslavia, Southern Europe, Northern/Western Europe, Predominantly Islamic countries (excluding Turkey), Asia and Other). No information was provided on country of birth, socioeconomic status, previous experience of mental illness, or other factors. While migrants reported higher rates of self-harm, there were differences based on where they were born. Students who had migrated from Poland or Southern Europe displayed the highest prevalence (30.8% and 25.8%, respectively), whereas students from other regions (e.g., Asia, Turkey, or predominantly Islamic countries) reported a lower 12-month prevalence of self-harm (13.3%, 15% and 14.9%, respectively) in comparison with German adolescents without a migration background 17.2%). The third study found that 19.2% of non-migrants engaged in self-harm, while between 36% and 66.7% of migrants had [48]. The study also reported that there are differences in self-harm based on country of origin. The data suggested that two-thirds (66.7%) of migrants born in Turkey and 60% of migrants born in Russia or the former Soviet Union engaged in self-harm, compared to 36.8% of migrants born in other countries. Besides Turkey and Russia, all other migrants were grouped as “other countries”, with no information distinguishing what country from which they originated. 

Two studies did not compare migrants with non-migrants. Out of these, one study which looked solely at young refugee children living in detention found that 25% had engaged in self-harm during their time in isolation [51]. Granular information on how long each individual had spent in isolation was not provided. Another study found that 17.4% of young migrants had a lifetime prevalence of self-harm, while 11.4% of young migrants engaged in self-harm in a 12-month period [54]. The study also provided a breakdown of methods of self-harm among young migrants noting that people may use more than one method. Fifty-five percent of young migrants who engaged in self-harm had used scratching as a method, 40% had banged or punched objects, 30% had banged or punched themselves, 25% used cutting and carving, and 20% had used burning as a method. The study found that of those who engaged in self-harm, 68.4% reported doing so more than five times. The study also looked at selected risk and protective factors, finding that gender, whether the migrant was accompanied or unaccompanied, living with or without both parents, and living in an asylum centre, had no significant influence on rates of self-harm. 

#### 3.4.3. Suicide Attempt

Eight studies reported data on attempted suicide, one of which obtained data by analysing the results of a national registry of data [45], while the remainder undertook surveys to ascertain self-reported episodes of attempted suicide [39,42,44,46,48,49,52]. Two studies only provided data from young migrants without comparing them to the host country’s non-migrant population [44,52], while the other studies compared migrant and non-migrant populations. Out of the studies that compared young migrants to non-migrants, one study found that young migrants have lower rates of suicide attempt (1%) compared to non-migrants (9%) [39]. The remaining five studies found that young migrants have higher rates of attempted suicide. 

A study in Germany found that 14.8% of young migrants from Poland or Southern Europe and 14.5% of young migrants from predominantly Islamic countries (except for Turkey) reported attempting suicide [39]. Young migrants from the former Yugoslavia reported almost the same prevalence (7.2%) as adolescents without a migration background (6.8%). The study also found that male students with a migration background reported a significantly higher lifetime prevalence of suicide attempts than native males (4.7% vs. 3.1%). 

Another study analysed data from multiple national registers (the Swedish Population and Housing Census, the Swedish National Parent Register, and the Register of the Total Swedish Population) for the year 1985 [45]. It found that 0.4% of immigrant boys and 1.9% of immigrant girls attempted suicide, compared to 0.3% in non-immigrant boys and 0.8% in non-immigrant girls. After adjustment for socioeconomic factors and parental risk factors, children in the adoptee group were 3.6 times more likely to die by suicide than those in the general population. The study also found that adoptees were more likely to have had psychiatric hospital care than non-adopted siblings. Another study found that 6.9% of first-generation migrants from Europe had attempted suicide, while 9.0% of non-European migrants had [46]. This was higher than non-migrants, 3.1% of who had attempted suicide. 

One study found that 17.9% of migrants had attempted suicide, compared to 3.2% of non-migrants [48]. Among migrants, 66.7% of those born in Turkey had attempted suicide, compared to 20% of those born in Russia and 10% of those born in other countries. Although the study looked at non-suicidal self-injury (NSSI) and suicide attempt, there was no way to determine a link between the two. Therefore, it was not possible to assess whether the same migrants who engaged in self-harm had also attempted suicide, or whether migrants reported engaging in these activities separately. Although one study grouped immigrants into those born in Asia or Latin America, without breaking them down into individual countries of origin [45], neither study provided granular information about the exact country of origin, gender, age at migration, and experience of mental illness and trauma, to be able to determine whether other confounding variables may explain the reasons for this difference. There was also no information presented about reasons for migration, suicide attempts prior to migration, diagnosis of any mental illness, or other such information.

Another study found that 10.4% of immigrant respondents had attempted suicide, compared to 4.4% of non-migrants in Russia and 8.7% of non-migrant in Israel [49]. As discussed previously, the study also reported data on suicidal ideation. Of the immigrant respondents who thought about suicide, 95.4% reported that they had attempted suicide. There were no data provided for non-migrants. There was also no information presented about reasons for migration, suicide attempts prior to migration, diagnosis of any mental illness, or other such information. Of the two studies that did not provide a comparison group of non-migrants, one found that 1.4% of young migrants attempt suicide [44], while the other found that 3.4% do [52]. Both studies looked specifically at young refugees. One study focused on children living in refugee camps in Greece and assessed a variety of health conditions and not just suicidal behaviour, including upper respiratory tract infections and dental hygiene [44]. The other study presented the results of a psychiatric epidemiological survey of adolescents from refugee families from 35 different countries. Neither study provided granular information on differences in rates of attempted suicide between refugees from different countries, information on previous diagnosis of mental illness, or family socioeconomic status. 

#### 3.4.4. Suicide Death

Only one study reported on suicide death, which was obtained by analysing data from Swedish national registers (Swedish Population and Housing Census of 1985, the Swedish National Parent Register, and the Register of the Total Swedish Population) for the cohort of people born in 1970–79 [45]. The study provided data on three different groups aged between 14 and 24 years—intercountry adoptees, immigrants, and non-migrants. Intercountry adoptees were defined as those born outside Europe, had immigrated to Sweden before 7 years of age, and had two parents who were born in Sweden without any record of emigration or immigration after 1968. Immigrants were defined as children who were born in Latin America or Asia and had arrived in Sweden before their 70th birthday but were recorded to have a mother who was born in their continent of origin. The main comparison group (non-migrants) consisted of children born in Sweden to two Swedish-born parents who had not emigrated or immigrated after 1968. After adjustment for socioeconomic factors and parental risk factors, children in the adoptee group were 3.6 times more likely to die by suicide than those in the general population. 

## 4. Discussion

This review examined 17 studies, the majority of which were published post 2011. Twelve studies provided data on suicidal ideation, five provided data on self-harm, eight provided data on suicide attempt, and one study included data on suicide death. 

### 4.1. Key Findings

This review has found that research into suicidal ideation and behaviour among young migrants has been limited and has produced mixed results. The findings suggest that suicidal ideation and behaviour in young migrants is complex, and it is difficult to delineate a theoretical framework explaining findings from the literature. Some studies indicate that young migrants experience higher rates of suicidal ideation compared to young people in the host population. Several authors have also suggested that suicide risk may vary among ethnic minorities, and they may have different and more specific risk factors for suicidal behaviour than the general population, such as acculturative stress. Conversely, other research has demonstrated that young people in the host population experience higher rates of suicidal ideation when compared to new migrants [39,41,47]. The lack of uniform findings reflects the complexity and heterogeneity of the term ‘migrants. Given migrants are comprised of different population groups, it is inappropriate to categorise them as one—which obscures the differences that exist between these groups. 

The Iceberg Model of Self-Harm posits that suicide represents only a fraction of the distress experienced by young people [55]. Research from the United Kingdom found that for every suicide death, 370 adolescents presented to hospital for self-harm and 3900 adolescents reported self-harm in the community [56]. This indicates that for each suicide death reported amongst young migrants, there are likely more young migrants engaging in self-harm, and many more still experiencing suicidal thoughts. Suicidal ideation and self-harm, besides being distressing in their own right, are also important predictors of suicide. Understanding the risk factors that result in suicidal behaviour properly is the first step needed to develop effective interventions to reduce suicidal ideation, self-harm, and suicidal behaviour. 

Rates of self-harm are higher among young migrants compared to non-migrants [41,42,48,54]. There appears to be high variance in rates of self-harm, with between 10% and 60% of young migrants having self-harmed at some point since migration. Young migrants also attempt suicide at higher rates than non-migrants. Up to two in three young migrants report having attempted suicide, compared to less than 1 in 10 non-migrants [42,45,46,48,49]. On the other hand, one study found that young international students attempt suicide less than their non-migrant counterparts [39]. It is unclear whether this is because these international students may belong to privileged socioeconomic groups with greater access to resources compared to university students in the host population who come from a variety of socioeconomic backgrounds. It is possible that had the study focused on young people in general, not just university students, they may have resulted in different findings. To understand the burden of suicide attempt, it is important to control for confounding variables and study young people from a variety of settings, not just university students. This highlights the need for research controlling for confounding variables such as socioeconomic status. Considered in combination with the findings that there are no major differences in suicidal ideation, these findings may suggest that young migrants do not significantly contemplate suicide before attempting it, implying greater impulsivity. It might also reflect issues of stigma and feeling too embarrassed to seek help, or the lack of culturally-sensitive mental health care services. Further research is required to understand whether there are reasons for why young migrants do not report high rates of suicidal ideation but attempt suicide more often. On the other hand, it is also possible that young migrants do not accurately understand the definition of suicidal ideation, and therefore do not report it correctly in studies. 

Young migrants from specific geographic regions are more likely to engage in some forms of suicidal ideation and/or behaviour compared to those from other regions. For example, a study posited that young migrants from Asia are more likely to have suicidal thoughts compared to migrants from Poland, North/Western Europe, and Southern Europe. However, young migrants from Poland, North/Western Europe, and Southern Europe have a higher prevalence of attempted suicide and self-harm compared to young Asian migrants [42]. This indicates that suicidal ideation may not be an accurate predictor of self-harm or attempted suicide in this population. Similarly, another study highlighted that young migrants to Western Europe from Turkey and the former Soviet Union had higher rates of self-harm than those born in other countries [48]. 

There does not appear to be any differences in suicide death between young migrants and non-migrants [45]. This finding suggests that regardless of whether young migrants have higher rates of self-harm or attempts at suicide, this does not necessarily result in a higher burden of death among this population. It is difficult to draw conclusions about this since only one study reported on suicide death, however, further research is warranted. 

From critically assessing the studies included in this review, some findings have emerged to guide the design of future studies. It is important to specifically focus on the country of origin, the country of destination, time spent in the country, reason for migration, previous diagnosis of any mental illness or suicidal behaviour prior to migration, and other confounding variables. It is also important to keep in mind that these factors may influence how migrants respond to questions. Besides the experiences after arrival in their host country, the mental health of migrants may be influenced by pre-migratory circumstances, such as reasons and motivation for migration and trauma experienced before and during the migration process [57]. Similarly, more acculturated young migrants may have more opportunities to interact with mainstream society, which results in greater stress from acculturation experiences and ultimately negatively impacts their mental health [58]. Another possible explanation is that the longer migrants spend in their host country, the more likely it is that they become more honest about expressing their feelings of emotional crisis such as suicidal ideation [59]. This indicates that young migrants may initially not honestly respond to questions around mental wellbeing, such as suicidal ideation, and become more honest the longer they have spent time in the host country. This would be particularly pronounced for migrants arriving from a country where the dominant culture values group harmony. For example, in some Asian countries, expressing thoughts such as suicidal ideation negatively affects group harmony and Asians are discouraged from expressing suicidal feelings [60,61]. This highlights the importance of assessing subjective cultural biases, which impact on the reporting of data. 

### 4.2. Limitations of the Included Studies

Based on the National Health Lung and Blood Institute’s Quality Assessment Tool for Observation Cohort and Cross-Sectional Studies, all studies included in this systematic review generally received a quality score of either “Fair” or “Poor”, with none of them obtaining a score of “Good”. This may reflect the difficulty when measuring a multi-faceted and complex issue such as migration, with numerous confounding variables that influence the participants before, during, and after the migration event. 

One of the challenges with some of the studies included in this review is that they grouped migrants into regions of origin rather than countries. This is problematic because each of these regions is comprised of diverse countries with different socioeconomic statuses. For example, referring to “predominantly Islamic countries” groups together migrants from low-income and/or war-torn countries who may have experienced trauma prior to and during migration with those from high-income countries (such as the United Arab Emirates and Kingdom of Saudi Arabia) who may not have these experiences [42]. It is important for studies to provide a breakdown of the country of origin, so that each country can be mapped to socioeconomic status, crime rates, education levels, and other factors to understand whether a young migrant’s country of birth affects their suicidal behaviour after migration. 

One of the limitations of conducting a review looking at suicidal behaviour and migration is the heterogeneity of study methodology. This is because each included study looked at different geographic locations where migrants originated from and migrated to. Furthermore, the studies did not specify reasons for migration. The studies also did not include historical information about suicidal behaviour, such as previous self-harm or attempted suicide prior to migration. Factors such as country of origin and destination, along with the reasons for migration and previous suicidal behaviour may explain suicidal behaviour after migration. To be truly insightful and enable comparison between studies, it would be beneficial for all included studies to capture this information so that the information can be compared with each other. 

All the studies included in this review were non-randomised control trials. Although we acknowledge the high risk of bias associated with non-randomised study designs, ethical considerations often prevent researchers studying suicidal behaviour from conducting RCTs. The original intention of the authors prior to commencing the study was to conduct a meta-analysis of the results. After discussion between the authors, it was decided that a meta-analysis would not be appropriate due to the heterogeneity in the outcome measures used between studies, limiting our ability to be confident that studies measured the same constructs. For example, methods to assess self-harm and suicidal ideation included a mixture of self-report instruments, hospital data, and parent and guardian-reported data. It was also often unclear if measures had been validated among young people. Researchers have called for the use of well-validated and standardised measures in suicide research, and we believe that the same is required in studies with young migrants [62]. 

### 4.3. Limitations of This Review

A key limitation of this review is that it focuses on the prevalence and epidemiology of suicidal behaviour. However, it is important to understand the risk and protective factors for these behaviours. Another limitation of the review is that it did not limit the country of origin and destination. Therefore, studies were included in this review where young people moved from one low socio-economic status country to another or from a high socioeconomic country to another. Given the country of origin plays a key role in determining the individual’s original circumstances, it is possible that the results would vary if an individual were to migrate from a low to high socioeconomic country, or vice versa. Some research also indicates that different ethnic groups exhibit different levels of suicidal behaviour [41,42]. It is important to explore these differences by categorising young migrants to clearly understand if differences exist. To reduce confounding, it is important to categorise countries of origin and destination appropriately, alongside ethnic background, to account for changes in socioeconomic and cultural environments. 

This review included studies that looked at both refugees and non-refugees. Refugees generally move to escape their country of origin and therefore have different motivations and lived experiences of trauma compared to individuals that have moved for employment or education opportunities [63]. It is important to account for the motivations of migration, which can be achieved by studying refugees and non-refugees separately. Furthermore, this review included studies that focused on only young migrants and therefore did not compare rates with the host population. These studies do not support this review’s aim of understanding whether there are differences in suicidal behaviour among migrants and non-migrants. To provide true insight, it is important to present comparable data across the two population groups. A final limitation is that we included studies that did not provide a non-migrant comparison group. Three studies provided findings of suicidal ideation from young migrants only and did not present any information about non-migrants [38,40,43]. Due to the unavailability of any control groups (i.e., non-migrants), it was not possible to draw any insights from these findings. This highlights the importance of ensuring that studies that analyse data about suicidal behaviour among young migrants should also collect and analyse data from non-migrants, either from the country of origin or from the country of destination. 

### 4.4. Implications

Notwithstanding the limitations above, this review has found that there appear to be differences in suicidal behaviour among young migrants and non-migrants. While most studies demonstrated that young migrants have higher rates of self-harm and suicide attempts compared to non-migrants, there were a few studies that found the opposite. Furthermore, it appears that there are no differences in suicidal ideation and suicide death when comparing young-migrants and non-migrants. The inconsistent results reflect a lack of knowledge and awareness about the lived experience of this vulnerable group of people. It is important to understand the risk and protective factors that result in engaging in suicidal behaviour, such as acculturation stress and isolation, coupled with barriers to accessing care due to stigma and shame, language barriers, and lack of culturally appropriate health service availability. Research is required to shed light and enhance knowledge of the factors that have a positive and negative effect on the individual. This is particularly relevant in this age of increasing migration, including by young migrants pursuing employment and educational opportunities and attempting to leave behind the impact of climate emergencies and conflict in their country of origin. This understanding is critical for policymakers, researchers and healthcare providers and will enable the development of policies and interventions that enhance the positive effects and reduce the negative effects of migration, enabling young migrants to enjoy a higher quality of life and reducing the risk of suicidal behaviour amongst this cohort. 

## 5. Conclusions

The findings of this review have highlighted that there are some differences in suicidal behaviour among young migrants compared to non-migrants. Research has indicated that young migrants have higher rates of self-harm and suicide attempt, but no major differences in suicidal ideation or suicide death. The inconsistent findings, and the limited number of studies that have focused on young migrants, indicates that further research is necessary to understand if there are any differences in suicidal behaviour among young migrants and non-migrants, and understand the risk and protective factors that lead to differences if they exist. 

## Figures and Tables

**Figure 1 ijerph-19-08329-f001:**
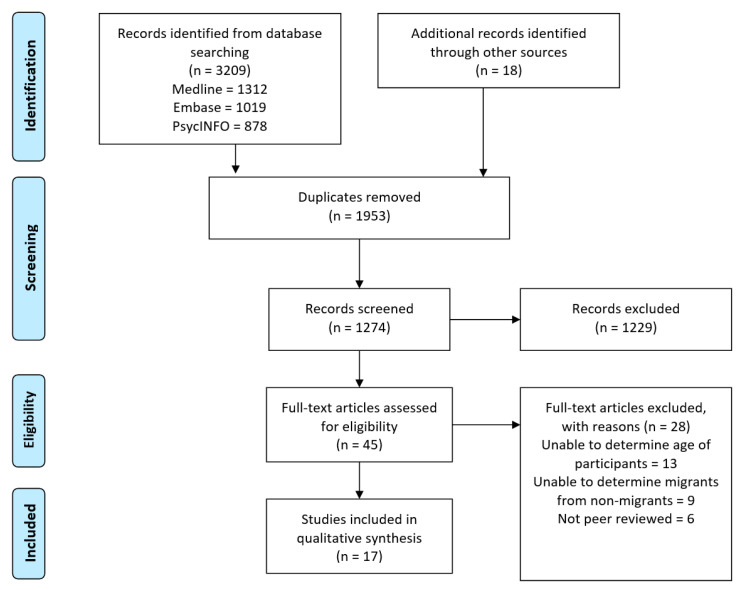
Results of PRISMA flow diagram.

**Table 1 ijerph-19-08329-t001:** Characteristics of Included Studies.

ID	Author and Year	N (Sample Size)	Country/Region of Birth	Country of Destination	Study Design	Method Used to Collect Data	Outcomes Assessed and Method of Reporting
1	Al-Maskari, F., Shah, S. M., Al-Sharhan, R., Al-Haj, E., Al-Kaabi, K., Khonji, D., Schneider, J. D., Nagelkerke, N. J. & Bernsen, R. M. (2011) [38]	318	India, Bangladesh, Pakistan, Non-national Arab countries and Others	United Arab Emirates	Case-control study	Survey of labour camps	Suicidal ideation (self-reported)
2	Aubert, P., Daigle, M. S., & Daigle, J. G. (2004) [39]	170	China	Canada	Case-control study	Survey of university students	Suicidal ideation; Attempted suicide (self-reported)
3	Betancourt, T. S., Newnham, E. A., Layne, C. M., Kim, S., Steinberg, A. M., Ellis, H & Birman, D. (2012) [40]	60	Central and South America, Africa, Eastern Europe, Asia, Middle East	United States of America	Case-control study	Analysis of data from mental health clinic	Suicidal ideation (self-reported)
4	Borges, G., Azrael, D., Almeida, J., Johnson, R. M., Molnar, B. E., Hemenway, D. & Miller, M. (2011) [41]	259	Numerous (not specified)	United States of America	Case-control study	Survey of high school students	Suicidal ideation; Self-harm (self-reported)
5	Donath, C., Bergmann, M. C., Kliem, S., Hillemacher, T. & Baier, D. (2019) [42]	10,638	Former Soviet Union, Turkey, Poland, Former Yugoslavia, Southern Europe, Northern/Western Europe, predominantly Islamic countries, Asia, Other countries.	Germany	Case-control study	Survey of high school students	Suicidal ideation; Self-harm; Attempted suicide (self-reported)
6	Elamoshy, R. & Feng, C. (2018) [43]	12,686	N/A	Canada	Case-control study	Survey of general population	Suicidal ideation (self-reported)
7	Hermans, M. P., Kooistra, J., Cannegieter, S. C., Rosendaal, F. R., Mook-Kanamori, D. O., & Nemeth, B. (2017) [44]	2291	Various	Greece	Cross-sectional study	Survey of refugees seeking clinical help	Suicide attempt (self-reported)
8	Hjern, A., Lindblad, F. & Vinnerljung, B. (2002) [45]	Total migrants (adoptees and immigrants) = 15,326 Intercountry adoptees = 11,320 Immigrants = 4006 Total 853,419	Various	Sweden	Case-control study	Analysis of national registry results	Attempted suicide; Suicide death (data from Swedish Hospital Discharge Register)
9	McMahon, E. M., Corcoran, P., Keeley, H., Cannon, M., Carli, V., Wasserman, C., Sarchiapone, M., Apter, A., Balazs, J., Banzer, R., Bobes, J., Brunner, R., Cozman, D., Haring, C., Kaess, M., Kahn, J. P., Kereszteny, A., Bitenc, U. M., Postuvan, V., Nemes, B., Saiz, P. A., Sisask, M., Tubiana, A., Varnik, P., Hoven, C. W. & Wasserman, D. (2017) [46]	Total = 11,057 Migrants = 663 First-generation migrants European origin = 428 First-generation migrants Non-European origin = 230 Non-migrants = 9018	Austria, Estonia, France, Germany, Hungary, Ireland, Italy, Romania, Slovenia, Spain	Europe	Case-control study	Survey of high school students	Suicidal ideation; Attempted suicide (self-reported)
10	Park, S. Y. (2019) [47]	Total = 9787 Asian American = 1418 Non-Hispanic White = 8369	Asia	United States of America	Case-control study	National survey	Suicidal ideation (self-reported)
11	Plener, P. L., Munz, L. M., Allroggen, M., Kapusta, N. D., Fegert, J. M. & Groschwitz, R. C. (2015) [48]	452	Turkey, Russia, Others	Germany	Case-control study	Survey of high school students	Self-harm; Attempted suicide (self-reported)
12	Ponizovsky, A. M., Ritsner, M. S. & Modai, I. (1999) [49]	52,380	Former Soviet Union	Israel	Case-control study	Survey of general population	Suicidal ideation; Attempted suicide (self-reported)
13	Slodnjak, V., Kos, A., & Yule, W. (2002) [50]	Refugee students = 265; Non-migrant children = 195	Bosnia and Herzegovina	Slovenia	Case-control study	Survey of high school students	Suicidal ideation (self-reported)
14	Steel, Z., Momartin, S., Bateman, C., Hafshejani, A., Silove, D.M., Everson, N., Roy, K., Dudley, M., Newman, L., Blick, B. and Mares, S. (2004) [51]	20 refugee children	Multiple	Australia	Cross-sectional study	Survey of refugees in detention	Suicidal ideation; Self-harm (self-reported)
15	Tousignant, M., Habimana, E., Biron, C., Malo, C., Sidoli-LeBlanc, E. and Bendris, N. (1999) [52]	203 refugee children	Multiple	Canada	Case-control study	Survey of refugee children; compared with data from school registry	Attempted suicide (self-reported)
16	Van Bergen, D. D., Smit, J. H., Van Balkom, A. J. L. M., Van Ameijden, E. & Saharso, S. (2008) [53]	203	Morocco, Turkey	The Netherlands	Case-control study	Analysis of registry data	Suicidal ideation (self-reported)
17	Verroken, S., Schotte, C., Derluyn, I. & Baetens, I. (2018) [54]	Total = 121 Asylum seeker = 31 Refugee = 40 Family reunion = 32 Other = 6	Various (Syria, Afghanistan, Iraq, Somalia, Other)	Belgium	Case-control study	Survey of high school students	Self-harm (self-reported)

## Data Availability

Not applicable.

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
