# Peer review of "Suicidal Behaviour, including Ideation and Self-Harm, in Young Migrants: A Systematic Review"

_ijerph, 2022, doi:10.3390/ijerph19148329_

Round 1

Reviewer 1 Report

Constructive feedback for authors:

I found your article to be quite insightful. Also, I believe that this topic is relevant in today’s socio-cultural climate and will add much value to the literature.

However, I wanted to provide some valuable feedback to help strengthen your paper. Please see below:

-Be mindful of language consistency. For instance, consider using death "by suicide" instead of death due to suicide

-Use gender neutral language “they” not his/her

- From the outset, be clear about the geographical location of this population.  Foe example, are you looking at this population globally or in a specific country?

- Lines 28-33 need to be restructured; long, run on, complex and fragmented sentence. make into concise and clear sentences.

- Materials and Methods (lines 73-75). Expand on this paragraph. Be explicit about the authors whose work informs your research. It might also be useful here to highlight the stages of the methodology for readers to help us dissert the process.

- Line 191 - Under outcome it might be useful to create a section that highlights themes that emerged from the study before addressing each in more depth to help with structure of your arguments and to guide the unfolding of your thoughts.

- Line 386 - might be useful to break findings down into themes with clear headings in order to provide a better understanding for readers of what these findings entail. The paragraph format is quite difficulted to follow and appears to lack coherency.

- Line 388 - young migrants from where? specify geographical regions.

- Line 400 - make stronger connection between iceberg model and research in the UK

- Line 470 Limitations of the included studies: use transitions to structure your ideas between paragraphs, which seems quite disjointed. Using topic sentences and be clear about limitations before addressing them will help with the flow of your ideas. This same feedback applies throughput the paper.

- Be clear in heading if these implications apply to researchers, policymakers, etc.

- Your target audience need to be more explicit in your writing. Who is this paper geared for?

Author Response

Dear reviewer, 

Thank you very much for providing your feedback.

We have actioned your changes in the updated version of the manuscript where possible. Here are my responses to your feedback: 

  1. Be mindful of language consistency. For instance, consider using death "by suicide" instead of death due to suicide. 
    Response: This article has used "death due to suicide" to remain consistent with other research articles. Some examples are: 
    - Wasserman, D., Cheng, Q., & Jiang, G. X. (2005). Global suicide rates among young people aged 15-19. World psychiatry : official journal of the World Psychiatric Association (WPA)4(2), 114–120. 
    - Hawton, K., Fagg, J., Platt, S., & Hawkins, M. (1993). Factors associated with suicide after parasuicide in young people. British Medical Journal, 306(6893), 1641-1644. 
    - Australian Institute of Health and Welfare 2021. The health impact of suicide and self-inflicted injuries in Australia, 2019. Cat. no. PHE 288.
    Canberra: AIHW. Viewed 10 November 2021, https://www.aihw.gov.au/reports/burden-of-disease/health-impact-suicide-self-inflictedinjuries-2019  

  2. Use gender neutral language “they” not his/her
    Response: Updated to make change. 

  3. From the outset, be clear about the geographical location of this population.  Foe example, are you looking at this population globally or in a specific country?
    Response:  Updated the Introduction and the Inclusion Criteria sub-section to make this clear. 

  4. Lines 28-33 need to be restructured; long, run on, complex and fragmented sentence. make into concise and clear sentences. 
    Response: Updated to simplify. 

  5. Materials and Methods (lines 73-75). Expand on this paragraph. Be explicit about the authors whose work informs your research. It might also be useful here to highlight the stages of the methodology for readers to help us dissert the process. 
    Response: Lines 61 - 62 highlight other systematic reviews that have been undertaken on related topics, which lists some of the authors whose work informed the research. The article follows the PRISMA protocol.

  6. Line 191 - Under outcome it might be useful to create a section that highlights themes that emerged from the study before addressing each in more depth to help with structure of your arguments and to guide the unfolding of your thoughts. 
    Response: It is normal when reporting systematic reviews to organise the results around outcomes of interest and that is what we have done here. In contrast we have to a certain extent structured the Discussion around the different themes that emerged from the data. 

  7. Line 386 - might be useful to break findings down into themes with clear headings in order to provide a better understanding for readers of what these findings entail. The paragraph format is quite difficulted to follow and appears to lack coherency. 
    Response: The entire Discussion section is structured around themes that have emerged from the included studies. 

  8. Line 388 - young migrants from where? specify geographical regions.
    Response: Updated the Introduction and the Inclusion Criteria sub-section to make this clear.

  9. Line 400 - make stronger connection between iceberg model and research in the UK. 
    Response: Whilst we appreciate the suggestion and think the Iceberg Model is extremely important, the intention of this review was not to make a commentary or link all findings to the Iceberg Model. 

  10. Line 470 Limitations of the included studies: use transitions to structure your ideas between paragraphs, which seems quite disjointed. Using topic sentences and be clear about limitations before addressing them will help with the flow of your ideas. This same feedback applies throughput the paper. 
    Response: The Discussion has been structured around themes and transitions have been made as consistently as possible. 

  11. Be clear in heading if these implications apply to researchers, policymakers, etc.
    Response: Updated last portion of Discussion to specify this. 

  12. Your target audience need to be more explicit in your writing. Who is this paper geared for?
    Response: This open-access article is geared for all readers, and not limited to any specific type. 

I hope that the above addresses your feedback to a satisfactory level. 

Kind regards, 

Aditya 

Reviewer 2 Report

Review of the paper: Suicidal Behaviour, Including Ideation and Self-Harm, in Young Migrants: A Systematic Review

The abstract of the article outline the background, methodology and the main results. In my opinion main contribution of the paper in the abstract might be more accentuated.

The paper feature a separate subchapter devoted to the description of the methodology.

The methological rigor of the article is at sufficient level. The authors did not formulate research hypotheses, but they are not necessary in the article.

The results are presented and commented with sound reasoning and appropriate interpretation.

Despite the undoubted merits of the article, I also have comments or questions that are worth answering:

-   I suggest you be clearer in your abstract about your research implications/limitations, and practical contribution;

-     in my opinion you should be more distinct about how your results contribute to theory and practice;

-     the question of whether other authors have done systematic review in this area of research is worth answering;

-   improve References: add page numbers (e. g. Ratkowska, K. A., & De Leo, D. Suicide in immigrants: An overview. Open Journal of Medical Psychology 2013, 2(3)).

Author Response

Dear reviewer, 

Thank you for providing your valuable feedback. 

Where required, we have made updates to the manuscript to action your comments. Here are my responses for each feedback item: 

  1. The abstract of the article outline the background, methodology and the main results. In my opinion main contribution of the paper in the abstract might be more accentuated. 
    Response: Updated the Abstract to add in an additional sentence making the contribution of the paper clearer. 

  2. The paper feature a separate subchapter devoted to the description of the methodology. 
    Response: The review has been completed following the PRISMA protocol as is best practice and is consistent with other systematic reviews published in this journal. We hope that this provides reassurance.

  3. The methological rigor of the article is at sufficient level. The authors did not formulate research hypotheses, but they are not necessary in the article.
    Response: No action required.

  4. The results are presented and commented with sound reasoning and appropriate interpretation.
    Response: No action required.

  5. I suggest you be clearer in your abstract about your research implications/limitations, and practical contribution. 
    Response: Updated Abstract to make this clearer.

  6. in my opinion you should be more distinct about how your results contribute to theory and practice. 
    Response: Added additional clarifying statements under the "Implications" sub-section.

  7. the question of whether other authors have done systematic review in this area of research is worth answering;
    Response: Other systematic reviews undertaken on related topics are acknowledged in Lines 64 - 65.
  8. improve References: add page numbers (e. g. Ratkowska, K. A., & De Leo, D. Suicide in immigrants: An overview. Open Journal of Medical Psychology 2013, 2(3)).
    Response: We have checked with the IJERPH Editorial Team who have evaluated the referencing and have advised no further action is needed at this stage. Additional work to make the referencing consistent will be undertaken in later steps, once the reviews are completed. 

I hope that the above addresses your feedback to a satisfactory level. 

Kind regards, 

Aditya 

Reviewer 3 Report

Thank you for the opportunity to review this paper. The authors conducted a systematic review of the literature examining suicidal behavior among young migrants. A total of 17 studies were included in this review, some of which reported on multiple outcomes of interest (e.g., suicidal ideation and self-harm). It was reported that the quality of the articles varied, but most reported that young migrants experience higher rates of self-harm and suicide attempt than young non-migrants. There was no difference in reported suicidal ideation or suicide death between young migrants and young non-migrants.

My feedback is as follows:

The Quality Assessment Tool is not a standardized measure of the quality of studies. Please justify your use of this tool in this study.

Given the low number of studies, I do wonder why the authors did not go beyond the systematic review and conduct a meta-analysis. Taking this approach would increase the overall rigor and quality of the analysis.

The study is fairly simple in design and conclusions drawn are very limited given the lack of studies available to include in the review. Although I can appreciate the highlighting of the need for additional research on suicidal behavior among this unique population, it is difficult to recommend publication of this paper due to the limitations of the findings. In other words, I am having trouble seeing a significant contribution to the literature through the publication of this paper. I am hopeful that the authors can expand on what they have developed to contribute further to the scant literature on this topic.

Author Response

Dear Reviewer, 

Thank you very much for providing your valuable feedback. 

Wherever possible, I have made changes to the updated version of the manuscript. Here are our responses for each of the feedback items: 

  1. The Quality Assessment Tool is not a standardized measure of the quality of studies. Please justify your use of this tool in this study.
    Response: Updated the "Study Quality" sub-section to clarify the tool. 

  2. Given the low number of studies, I do wonder why the authors did not go beyond the systematic review and conduct a meta-analysis. Taking this approach would increase the overall rigor and quality of the analysis.
    Response: A narrative synthesis of the results was undertaken to obtain insights from the data. Meta‐analysis was not possible due to the heterogeneity in the outcome measures used between studies and the variety of settings from which data were captured, limiting the ability to accurately compare results. Undertaking a meta analysis in this environment runs the risk of providing inaccurate results and so was not undertaken. This has been acknowledged in the "Data Synthesis" sub-section. 

  3. The study is fairly simple in design and conclusions drawn are very limited given the lack of studies available to include in the review. Although I can appreciate the highlighting of the need for additional research on suicidal behavior among this unique population, it is difficult to recommend publication of this paper due to the limitations of the findings. In other words, I am having trouble seeing a significant contribution to the literature through the publication of this paper. I am hopeful that the authors can expand on what they have developed to contribute further to the scant literature on this topic.
    Response: The limitations of the study have been acknowledged in the "Limitations of this review" sub-section. We believe that there is still merit in the study given the importance of the topic and that it still brings new information and this was reinforced by the other reviewers. Whilst we take your point, we need to be very cautious when making recommendations / conclusions as they can’t go beyond the data we have. 

I hope this addresses your feedback to a satisfactory level. 

Kind regards, 

Aditya 

Reviewer 4 Report

Thank you for the opportunity to review the manuscript, entitled "Suicidal Behaviour, Including Ideation and Self-Harm, in Young Migrants: A Systematic Review". This is a valuable and important work, especially timely performed during the current emigration crisis. The introduction is a well-written and concise review of current literature, which includes a necessary important information. The method is exhaustively described, allowing replication of the study. Results are presented clearly and transparently. Discussion and conclusions are the critical and purposeful elaboration of relevant data. However, some minor revisions can improve the study:

1. First of all, reference style (i.e., reference list and citation in text) is inconsistent with the MDPI guideline and need improvement in the full article.

2. The first sentence (page 1, lines 28-34) in the Introduction section is too long. I propose to present each definition in a separate sentence, apart from listing various forms of harmful behavior of adolescents and young adults.

3. Definition of young people and international migrant status should be presented in the Introduction section, instead of Methods (page 2, lines 81-88).

4. Appendix B is completely unavailable for the reader. Please include the data in the Supplementary material instead of Appendix.

Author Response

Dear Reviewer, 

Thank you very much for providing your valuable feedback. 

Where possible, we have updated the manuscript to address your comments. Here are my individual responses for each comment: 

  1. First of all, reference style (i.e., reference list and citation in text) is inconsistent with the MDPI guideline and need improvement in the full article.
    Response: We have consulted with the IJERPH Editorial Team who have reviewed the referencing and have stated that no action is required at this stage. Further work will be undertaken in the later stages to ensure consistency with the journal's requirements. 

  2. The first sentence (page 1, lines 28-34) in the Introduction section is too long. I propose to present each definition in a separate sentence, apart from listing various forms of harmful behavior of adolescents and young adults.
    Response: This has been updated.

  3. Definition of young people and international migrant status should be presented in the Introduction section, instead of Methods (page 2, lines 81-88). 
    Response: Updated to move the definition to the Introduction section.

  4. Appendix B is completely unavailable for the reader. Please include the data in the Supplementary material instead of Appendix.
    Response: This has now been updated as Supplementary material. 

I hope that these changes are satisfactory. 

Kind regards, 

Aditya 

Round 2

Reviewer 3 Report

Thank you for the opportunity to review the revised version of this paper. The authors conducted a systematic review of the literature examining suicidal behavior among young migrants. Revisions to the paper were minimal but concerns were sufficiently addressed. I also appreciate the authors’ acknowledgement of the study’s limitations. However, I still do struggle to see its added value to the current literature base. As a submission to a special issue on “Novel Approaches to Suicide Prevention,” I don’t see how this article fits. But as an article to be published outside of the special issue, I can see its merit and hope that it raises awareness of the need for further research on this topic.

Author Response

Dear reviewer, 

Thank you for your valuable feedback. 

The reason for this review being published in the “Novel Approaches to Suicide Prevention" special edition is because it highlights the lack of primary research available on suicidal behaviour among young migrants. Before designing any approaches to suicide prevention, the first step is to have a clear understanding of what the problem is. This review highlights that this understanding does not clearly exist and indicates the need for research to bridge this gap. This should then enable the development of suicide prevention interventions. The journal itself recommended that this review be included in this special edition. 

I hope that this addresses your query. 

Kind regards, 
Aditya